# FEDERATED LEARNING FROM SMALL DATASETS

**Michael Kamp**
Institute for AI in medicine (IKIM)
University Hospital Essen, Essen Germany, and
Ruhr-University Bochum, Bochum Germany, and
Monash University, Melbourne, Australia
michael.kamp@uk-essen.de

**Jonas Fischer**
Harvard T.H. Chan School of Public Health
Department of Biostatistics
Boston, MA, United States
jfischer@hsph.harvard.edu

**Jilles Vreeken**
CISPA Helmholtz Center for Information Security
Saarbrücken, Germany
vreeken@cispa.de

## ABSTRACT

Federated learning allows multiple parties to collaboratively train a joint model without having to share any local data. It enables applications of machine learning in settings where data is inherently distributed and undisclosable, such as in the medical domain. Joint training is usually achieved by aggregating local models. When local datasets are small, locally trained models can vary greatly from a globally good model. Bad local models can arbitrarily deteriorate the aggregate model quality, causing federating learning to fail in these settings. We propose a novel approach that avoids this problem by interleaving model *aggregation* and *permutation* steps. During a permutation step we redistribute local models across clients through the server, while preserving data privacy, to allow each local model to train on a daisy chain of local datasets. This enables successful training in data-sparse domains. Combined with model aggregation, this approach enables effective learning even if the local datasets are extremely small, while retaining the privacy benefits of federated learning.

## 1 INTRODUCTION

How can we learn *high quality* models when data is *inherently distributed* across sites and cannot be shared or pooled? In federated learning, the solution is to iteratively train models locally at each site and share these models with the server to be aggregated to a global model. As only models are shared, data usually remains undisclosed. This process, however, requires sufficient data to be available at each site in order for the locally trained models to achieve a minimum quality—even a single bad model can render aggregation arbitrarily bad (Shamir and Srebro, 2014). In many relevant applications this requirement is not met: In healthcare settings we often have as little as a few dozens of samples (Granlund et al., 2020; Su et al., 2021; Painter et al., 2020). Also in domains where deep learning is generally regarded as highly successful, such as natural language processing and object detection, applications often suffer from a lack of data (Liu et al., 2020; Kang et al., 2019).

To tackle this problem, we propose a new building block called *daisy-chaining* for federated learning in which models are trained on one local dataset after another, much like a daisy chain. In a nutshell, at each client a model is trained locally, sent to the server, and then—instead of aggregating local models—sent to a random other client as is (see Fig. 1). This way, each local model is exposed to a daisy chain of clients and their local datasets. This allows us to learn from small, distributed datasets simply by consecutively training the model with the data available at each site. Daisy-chaining alone, however, violates privacy, since a client can infer from a model upon the data of the client it received it from (Shokri et al., 2017). Moreover, performing daisy-chaining naively would lead to overfitting which can cause learning to diverge (Haddadpour and Mahdavi, 2019). In this paper, we propose to combine daisy-chaining of local datasets with aggregation of models, both orchestrated by the server, and term this method *federated daisy-chaining* (FEDDC).

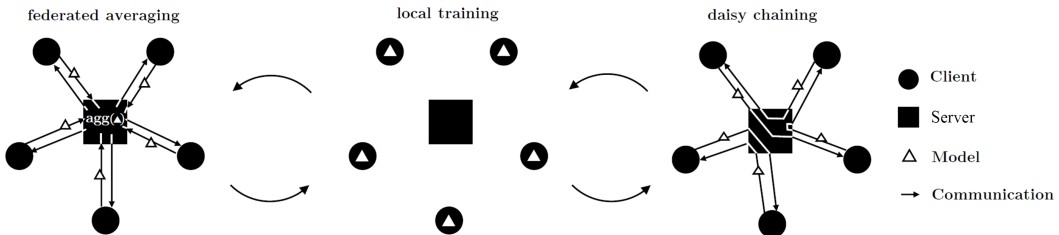

Figure 1: *Federated learning settings.* A standard federated learning setting with training of local models at clients (middle) with aggregation phases where models are communicated to the server, aggregated, and sent back to each client (left). We propose to add daisy chaining (right), where local models are sent to the server and then redistributed to a random permutation of clients as is.

We show that our simple, yet effective approach maintains privacy of local datasets, while it provably converges and guarantees improvement of model quality in convex problems with a suitable aggregation method. Formally, we show convergence for FEDDC on non-convex problems. We then show for convex problems that FEDDC succeeds on small datasets where standard federated learning fails. For that, we analyze FEDDC combined with aggregation via the Radon point from a PAC-learning perspective. We substantiate this theoretical analysis for convex problems by showing that FEDDC in practice matches the accuracy of a model trained on the full data of the SUSY binary classification dataset with only 2 samples per client, outperforming standard federated learning by a wide margin. For non-convex settings, we provide an extensive empirical evaluation, showing that FEDDC outperforms naive daisy-chaining, vanilla federated learning FEDAVG (McMahan et al., 2017), FEDPROX (Li et al., 2020a), FEDADAGRAD, FEDADAM, and FEDYOGI (Reddi et al., 2020) on low-sample CIFAR10 (Krizhevsky, 2009), including non-iid settings, and, more importantly, on two real-world medical imaging datasets. Not only does FEDDC provide a wide margin of improvement over existing federated methods, but it comes close to the performance of a gold-standard (centralized) neural network of the same architecture trained on the pooled data. To achieve that, it requires a small communication overhead compared to standard federated learning for the additional daisy-chaining rounds. As often found in healthcare, we consider a cross-SILO scenario where such small communication overhead is negligible. Moreover we show that with equal communication, standard federated averaging still underperforms in our considered settings.

In summary, our contributions are (i) FEDDC, a novel approach to federated learning from small datasets via a combination of model permutations across clients and aggregation, (ii) a formal proof of convergence for FEDDC, (iii) a theoretical guarantee that FEDDC improves models in terms of $\epsilon, \delta$-guarantees which standard federated learning can not, (iv) a discussion of the privacy aspects and mitigations suitable for FEDDC, including an empirical evaluation of differentially private FEDDC, and (v) an extensive set of experiments showing that FEDDC substantially improves model quality on small datasets compared to standard federated learning approaches.

## 2    RELATED WORK

Learning from small datasets is a well studied problem in machine learning. In the literature, we find both general solutions, such as using simpler models and transfer learning (Torrey and Shavlik, 2010), and more specialized ones, such as data augmentation (Ibrahim et al., 2021) and few-shot learning (Vinyals et al., 2016; Prabhu et al., 2019). In our scenario overall data is abundant, but the problem is that data is distributed into small local datasets at each site, which we are not allowed to pool. Hao et al. (2021) propose local data augmentation for federated learning, but their method requires a sufficient quality of the local model for augmentation which is the opposite of the scenario we are considering. Huang et al. (2021) provide generalization bounds for federated averaging via the NTK-framework, but requires one-layer infinite-width NNs and infinitesimal learning rates.

Federated learning and its variants have been shown to learn from incomplete local data sources, e.g., non-iid label distributions (Li et al., 2020a; Wang et al., 2019) and differing feature distributions (Li et al., 2020b; Reisizadeh et al., 2020a), but fail in case of large gradient diversity (Haddadpour and Mahdavi, 2019) and strongly dissimilar label distribution (Marfoq et al., 2021). For small

datasets, local empirical distributions may vary greatly from the global distribution: the difference of empirical to true distribution decreases exponentially with the sample size (e.g., according to the Dvoretzky–Kiefer–Wolfowitz inequality), but for small samples the difference can be substantial, in particular if the distribution differs from a Normal distribution (Kwak and Kim, 2017). Shamir and Srebro (2014) have shown the adverse effect of bad local models on averaging, proving that even due to a single bad model averaging can be arbitrarily bad.

A different approach to dealing with biased local data is by learning personalized models at each client. Such personalized FL (Li et al., 2021) can reduce sample complexity, e.g., by using shared representations (Collins et al., 2021) for client-specific models, e.g., in the medical domain (Yang et al., 2021), or by training sample-efficient personalized Bayesian methods (Achituve et al., 2021). It is not applicable, however, to settings where you are not allowed to learn the biases or batch effects of local clients, e.g., in many medical applications where this would expose sensitive client information. Kiss and Horvath (2021) propose a decentralized and communication-efficient variant of federated learning that migrates models over a decentralized network, storing incoming models locally at each client until sufficiently many models are collected on each client for an averaging step, similar to Gossip federated learing (Jelasity et al., 2005). The variant without averaging is similar to simple daisy-chaining which we compare to in Section 7. FEDDC is compatible with any aggregation operator, including the Radon machine (Kamp et al., 2017), the geometric median (Pillutla et al., 2022), or neuron-clustering (Yurochkin et al., 2019), and can be straightforwardly combined with approaches to improve communication-efficiency, such as dynamic averaging (Kamp et al., 2018), and model quantization (Reisizadeh et al., 2020b). We combine FEDDC with averaging, the Radon machine, and FedProx (Li et al., 2020a) in Sec. 7.

## 3 PRELIMINARIES

We assume iterative learning algorithms (cf. Chp. 2.1.4 Kamp, 2019) $\mathcal{A} : \mathcal{X} \times \mathcal{Y} \times \mathcal{H} \to \mathcal{H}$ that update a model $h \in \mathcal{H}$ using a dataset $D \subset \mathcal{X} \times \mathcal{Y}$ from an input space $\mathcal{X}$ and output space $\mathcal{Y}$, i.e., $h_{t+1} = \mathcal{A}(D, h_t)$. Given a set of $m \in \mathbb{N}$ clients with local datasets $D^1, \ldots, D^m \subset \mathcal{X} \times \mathcal{Y}$ drawn iid from a data distribution $\mathcal{D}$ and a loss function $\ell : \mathcal{Y} \times \mathcal{Y} \to \mathbb{R}$, the goal is to find a single model $h^* \in \mathcal{H}$ that minimizes the risk $\varepsilon(h) = \mathbb{E}_{(x,y) \sim \mathcal{D}}[\ell(h(x), y)]$. In *centralized learning*, datasets are pooled as $D = \bigcup_{i \in [m]} D^i$ and $\mathcal{A}$ is applied to $D$ until convergence. Note that applying $\mathcal{A}$ on $D$ can be the application to any random subset, e.g., as in mini-batch training, and convergence is measured in terms of low training loss, small gradient, or small deviation from previous iterate. In standard *federated learning* (McMahan et al., 2017), $\mathcal{A}$ is applied in parallel for $b \in \mathbb{N}$ rounds on each client locally to produce local models $h^1, \ldots, h^m$. These models are then centralized and aggregated using an aggregation operator $\mathrm{agg} : \mathcal{H}^m \to \mathcal{H}$, i.e., $\overline{h} = \mathrm{agg}(h^1, \ldots, h^m)$. The aggregated model $\overline{h}$ is then redistributed to local clients which perform another $b$ rounds of training using $\overline{h}$ as a starting point. This is iterated until convergence of $\overline{h}$. When aggregating by averaging, this method is known as federated averaging (FEDAVG). Next, we describe FEDDC.

## 4 FEDERATED DAISY-CHAINING

We propose federated daisy chaining as an extension to federated learning in a setup with $m$ clients and one designated sever.[1] We provide the pseudocode of our approach as Algorithm 1.

**The client:** Each client trains its local model in each round on local data (line 4), and sends its model to the server every $b$ rounds for aggregation, where $b$ is the aggregation period, and every $d$ rounds for daisy chaining, where $d$ is the daisy-chaining period (line 6). This re-distribution of models results in each individual model conceptually following a daisy chain of clients, training on each local dataset. Such a daisy chain is interrupted by each aggregation round.

**The server:** Upon receiving models, in a daisy-chaining round (line 9) the server draws a random permutation $\pi$ of clients (line 10) and re-distributes the model of client $i$ to client $\pi(i)$ (line 11), while in an aggregation round (line 12), the server instead aggregates all local models and re-distributes the aggregate to all clients (line 13-14).

---

[1]This star-topology can be extended to hierarchical networks in a straightforward manner. Federated learning can also be performed in a decentralized network via gossip algorithms (Jelasity et al., 2005).

---

**Algorithm 1:** Federated Daisy-Chaining FEDDC

---

**Input:** daisy-chaining period $d$, aggregation period $b$, learning algorithm $\mathcal{A}$, aggregation operator agg, $m$ clients with local datasets $D^1, \ldots, D^m$, total number of rounds $T$

**Output:** final model aggregate $\overline{h}_T$

1   initialize local models $h_0^1, \ldots, h_0^m$
2   **Locally** *at client $i$ at time $t$* **do**
3      sample $S$ from $D^i$
4      $h_t^i \leftarrow \mathcal{A}(S, h_{t-1}^i)$
5      **if** $t \bmod d = d - 1$ *or* $t \bmod b = b - 1$ **then**
6         send $h_t^i$ to server
7         receive new $h_t^i$ from server               // receives either aggregate $\overline{h}_t$ or some $h_t^j$
8   **At server** *at time $t$* **do**
9      **if** $t \bmod d = d - 1$ **then**                             // daisy chaining
10        draw permutation $\pi$ of [1,m] at random
11        for all $i \in [m]$ send model $h_t^i$ to client $\pi(i)$
12      **else if** $t \bmod b = b - 1$ **then**                    // aggregation
13        $\overline{h}_t \leftarrow \text{agg}(h_t^1, \ldots, h_t^m)$
14        send $\overline{h}_t$ to all clients

---

**Communication complexity:** Note that we consider cross-SILO settings, such as healthcare, were communication is not a bottleneck and, hence, restrict ourselves to a brief discussion in the interest of space. Communication between clients and server happens in $O(\frac{T}{d} + \frac{T}{b})$ many rounds, where $T$ is the overall number of rounds. Since FEDDC communicates every $d$th and $b$th round, the amount of communication rounds is similar to FEDAVG with averaging period $b_{FedAvg} = \min\{d, b\}$. That is, FEDDC increases communication over FEDAVG by a constant factor depending on the setting of $b$ and $d$. The amount of communication per communication round is linear in the number of clients and model size, similar to federated averaging. We investigate the performance of FEDAVG provided with the same communication capacity as FEDDC in our experiments and in App. A.3.6.

## 5   THEORETICAL GUARANTEES

In this section, we formally show that FEDDC converges for averaging. We, further, provide theoretical bounds on the model quality in convex settings, showing that FEDDC has favorable generalization error in low sample settings compared to standard federated learning. More formally, we first show that under standard assumptions on the empirical risk, it follows from a result of Yu et al. (2019) that FEDDC converges when using averaging as aggregation and SGD for learning—a standard setting in, e.g., federated learning of neural networks. We provide all proofs in the appendix.

**Corollary 1.** *Let the empirical risks $\mathcal{E}_{emp}^i(h) = \sum_{(x,y) \in D^i} \ell(h_i(x), y)$ at each client $i \in [m]$ be $L$-smooth with $\sigma^2$-bounded gradient variance and $G^2$-bounded second moments, then FEDDC with averaging and SGD has a convergence rate of $\mathcal{O}(1/\sqrt{mT})$, where $T$ is the number of local updates.*

Since model quality in terms of generalization error does not necessarily depend on convergence of training (Haddadpour and Mahdavi, 2019; Kamp et al., 2018), we additionally analyze model quality in terms of probabilistic worst-case guarantees on the generalization error (Shalev-Shwartz and Ben-David, 2014). The average of local models can yield as bad a generalization error as the worst local model, hence, using averaging as aggregation scheme in standard federated learning can yield arbitrarily bad results (cf. Shamir and Srebro, 2014). As the probability of bad local models starkly increases with smaller sample sizes, this trivial bound often carries over to our considered practical settings. The Radon machine (Kamp et al., 2017) is a federated learning approach that overcomes these issues for a wide range of learning algorithms and allows us to analyze (non-trivial) quality bounds of aggregated models under the assumption of convexity. Next, we show that FEDDC can improve model quality for small local datasets where standard federated learning fails to do so.

A Radon point (Radon, 1921) of a set of points $S$ from a space $\mathcal{X}$ is—similar to the geometric median—a point in the convex hull of $S$ with a high centrality (i.e., a Tukey depth (Tukey, 1975;

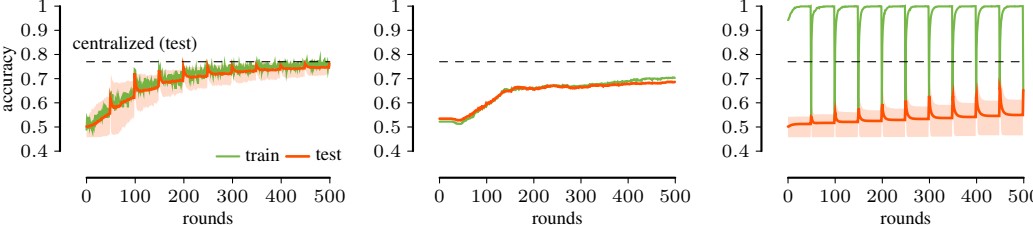

(a) FEDDC with Radon point with $d = 1$, $b = 50$.

(b) Federated learning with Radon point with $b = 1$.

(c) Federated learning with Radon point with $b = 50$.

Figure 2: *Results on SUSY*. We visualize results in terms of train (green) and test error (orange) for (a) FEDDC ($d = 1$, $b = 50$) and standard federated learning using Radon points for aggregation with (b) $b = 1$, i.e., the same amount of communication as FEDDC, and (c) $b = 50$, i.e., the same aggregation period as FEDDC. The network has 441 clients with 2 data points per client. The performance of a central model trained on all data is indicated by the dashed line.

Gilad-Bachrach et al., 2004) of at least 2). For a Radon point to exist, $S \subset \mathcal{X}$ has to have a minimum size $r \in \mathbb{N}$ called the Radon number of $\mathcal{X}$. For $\mathcal{X} \subseteq \mathbb{R}^d$ the radon number is $d + 2$. Here, the set of points $S$ are the local models, or more precisely their parameter vectors. We make the following standard assumption (Von Luxburg and Schölkopf, 2011) on the local learning algorithm $\mathcal{A}$.

**Assumption 2** (($\epsilon, \delta$)-guarantees). *The learning algorithm $\mathcal{A}$ applied on a dataset drawn iid from $\mathcal{D}$ of size $n \geq n_0 \in \mathbb{N}$ produces a model $h \in \mathcal{H}$ s.t. with probability $\delta \in (0, 1]$ it holds for $\epsilon > 0$ that $\mathbb{P}\left(\varepsilon(h) > \epsilon\right) < \delta$. The sample size $n_0$ is monotonically decreasing in $\delta$ and $\epsilon$ (note that typically $n_0$ is a polynomial in $\epsilon^{-1}$ and $\log(\delta^{-1})$).*

Here $\varepsilon(h)$ is the risk defined in Sec. 3. Now let $r \in \mathbb{N}$ be the Radon number of $\mathcal{H}$, $\mathcal{A}$ be a learning algorithm as in assumption 2, and risk $\varepsilon$ be convex. Assume $m \geq r^h$ many clients with $h \in \mathbb{N}$. For $\epsilon > 0, \delta \in (0, 1]$ assume local datasets $D_1, \ldots, D_m$ of size larger than $n_0(\epsilon, \delta)$ drawn iid from $\mathcal{D}$, and $h_1, \ldots, h_m$ be local models trained on them using $\mathcal{A}$. Let $\mathfrak{r}_h$ be the iterated Radon point (Clarkson et al., 1996) with $h$ iterations computed on the local models (for details, see App. A.2). Then it follows from Theorem 3 in Kamp et al. (2017) that for all $i \in [m]$ it holds that

$$\mathbb{P}\left(\varepsilon(\mathfrak{r}_h) > \epsilon\right) \leq \left(r\,\mathbb{P}\left(\varepsilon(h_i) > \epsilon\right)\right)^{2^h} \tag{1}$$

where the probability is over the random draws of local datasets. That is, the probability that the aggregate $\mathfrak{r}_h$ is bad is doubly-exponentially smaller than the probability that a local model is bad. Note that in PAC-learning, the error bound and the probability of the bound to hold are typically linked, so that improving one can be translated to improving the other (Von Luxburg and Schölkopf, 2011). Eq. 1 implies that the iterated Radon point only improves the guarantee on the confidence compared to that for local models if $\delta < r^{-1}$, i.e. $\mathbb{P}\left(\varepsilon(\mathfrak{r}_h) > \epsilon\right) \leq \left(r\,\mathbb{P}\left(\varepsilon(h_i) > \epsilon\right)\right)^{2^h} < (r\delta)^{2^h} < 1$ only holds for $r\delta < 1$. Consequently, local models need to achieve a minimum quality for the federated learning system to improve model quality.

**Corollary 3.** *Let $\mathcal{H}$ be a model space with Radon number $r \in \mathbb{N}$, $\varepsilon$ a convex risk, and $\mathcal{A}$ a learning algorithm with sample size $n_0(\epsilon, \delta)$. Given $\epsilon > 0$ and any $h \in \mathbb{N}$, if local datasets $D_1, \ldots, D_m$ with $m \geq r^h$ are smaller than $n_0(\epsilon, r^{-1})$, then federated learning using the Radon point does not improve model quality in terms of ($\epsilon, \delta$)-guarantees.*

In other words, when using aggregation by Radon points alone, an improvement in terms of ($\epsilon, \delta$)-guarantees is strongly dependent on large enough local datasets. Furthermore, given $\delta > r^{-1}$, the guarantee can become arbitrarily bad by increasing the number of aggregation rounds.

Federated Daisy-Chaining as given in Alg. 1 permutes local models at random, which is in theory equivalent to permuting local datasets. Since the permutation is drawn at random, the amount of permutation rounds $T$ necessary for each model to observe a minimum number of distinct datasets $k$ with probability $1 - \rho$ can be given with high probability via a variation of the coupon collector problem as $T \geq d\frac{m}{\rho^{\frac{1}{m}}}(H_m - H_{m-k})$, where $H_m$ is the $m$-th harmonic number—see Lm. 5 in

App. A.5 for details. It follows that when we perform daisy-chaining with $m$ clients and local datasets of size $n$ for at least $dm\rho^{-\frac{1}{m}}(H_m - H_{m-k})$ rounds, then each local model will with probability at least $1 - \rho$ be trained on at least $kn$ distinct samples. For an $\epsilon, \delta$-guarantee, we thus need to set $b$ large enough so that $kn \geq n_0(\epsilon, \sqrt{\delta})$ with probability at least $1 - \sqrt{\delta}$. This way, the failure probability is the product of not all clients observing $k$ distinct datasets and the model having a risk larger than $\epsilon$, which is $\sqrt{\delta}\sqrt{\delta} = \delta$.

**Proposition 4.** *Let $\mathcal{H}$ be a model space with Radon number $r \in \mathbb{N}$, $\varepsilon$ a convex risk , and $\mathcal{A}$ a learning algorithm with sample size $n_0(\epsilon, \delta)$. Given $\epsilon > 0$, $\delta \in (0, r^{-1})$ and any $h \in \mathbb{N}$, and local datasets $D_1, \ldots, D_m$ of size $n \in \mathbb{N}$ with $m \geq r^h$, then Alg. 1 using the Radon point with aggr. period*

$$b \geq d\frac{m}{\delta^{\frac{1}{2m}}}\left(H_m - H_{m-\lceil n^{-1}n_0(\epsilon, \sqrt{\delta})\rceil}\right) \tag{2}$$

*improves model quality in terms of $(\epsilon, \delta)$-guarantees.*

This result implies that if enough daisy-chaining rounds are performed in-between aggregation rounds, federated learning via the iterated Radon point improves model quality in terms of $(\epsilon, \delta)$-guarantees: the resulting model has generalization error smaller than $\epsilon$ with probability at least $1 - \delta$. Note that the aggregation period cannot be arbitrarily increased without harming convergence. To illustrate the interplay between these variables, we provide a numerical analysis of Prop. 4 in App. A.5.1.

This theoretical result is also evident in practice, as we show in Fig. 2. There, we compare FEDDC with standard federated learning and equip both with the iterated Radon point on the SUSY binary classification dataset (Baldi et al., 2014). We train a linear model on 441 clients with only 2 samples per client. After 500 rounds FEDDC daisy-chaining every round ($d = 1$) and aggregating every fifty rounds ($b = 50$) reached the test accuracy of a gold-standard model that has been trained on the centralized dataset (ACC=0.77). Standard federated learning with the same communication complexity using $b = 1$ is outperformed by a large margin (ACC=0.68). We additionally provide results of standard federated learning with $b = 50$ (ACC=0.64), which shows that while the aggregated models perform reasonable, the standard approach heavily overfits on local datasets if not pulled to a global average in every round. More details on this experiment can be found in App. A.3.2. In Sec. 7 we show that the empirical results for averaging as aggregation operator are similar to those for the Radon machine. First, we discuss the privacy-aspects of FEDDC.

## 6 DATA PRIVACY

A major advantage of federated over centralized learning is that local data remains undisclosed to anyone but the local client, only model parameters are exchanged. This provides a natural benefit to data privacy, which is the main concern in applications such as healthcare. However, an attacker can make inferences about local data from model parameters (Ma et al., 2020) and model updates or gradients (Zhu and Han, 2020). In the daisy-chaining rounds of FEDDC clients receive a model that was directly trained on the local data of another client, instead of a model aggregate, potentially facilitating membership inference attacks (Shokri et al., 2017)—reconstruction attacks (Zhu and Han, 2020) remain difficult because model updates cannot be inferred since the server randomly permutes the order of clients in daisy-chaining rounds.

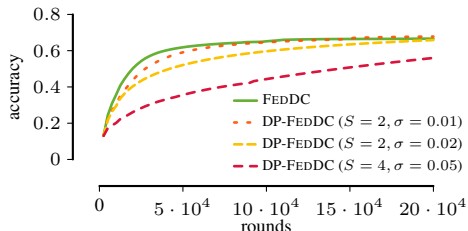

Figure 3: *Differential privacy results.* Comparison of FEDDC (top solid line) to FEDDC with clipped parameter updates and Gaussian noise (dashed lines) on CIFAR10 with 250 clients.

Should a malicious client obtain model updates through additional attacks, a common defense is applying appropriate clipping and noise before sending models. This guarantees $\epsilon, \delta$-differential privacy for local data (Wei et al., 2020) at the cost of a slight-to-moderate loss in model quality. This technique is also proven to defend against backdoor and poisoning attacks (Sun et al., 2019). Moreover, FEDDC is compatible with standard defenses against such attacks, such as noisy or robust aggregation (Liu et al., 2022)—FEDDC with the Radon machine is an example of robust aggregation. We illustrate the effectiveness of FEDDC

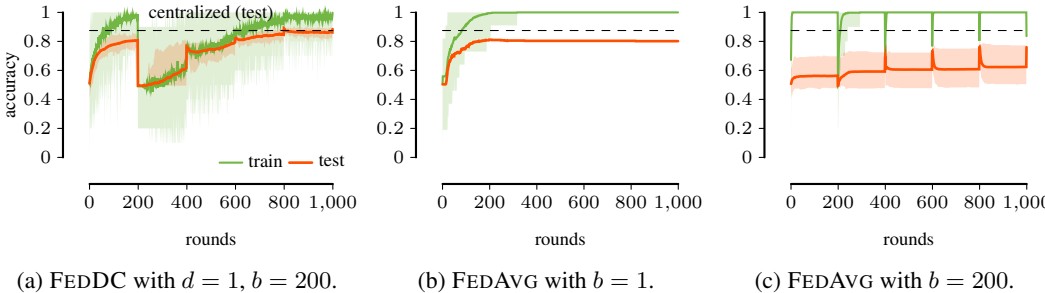

(a) FEDDC with $d = 1$, $b = 200$.  (b) FEDAVG with $b = 1$.  (c) FEDAVG with $b = 200$.

Figure 4: *Synthetic data results.* Comparison of FEDDC (a), FEDAVG with same communication (b) and same averaging period (c) for training fully connected NNs on synthetic data. We report mean and confidence accuracy per client in color and accuracy of central learning as dashed black line.

with differential privacy in the following experiment. We train a small ResNet on 250 clients using FEDDC with $d = 2$ and $b = 10$, postponing the details on the experimental setup to App. A.1.1 and A.1.2. Differential privacy is achieved by clipping local model updates and adding Gaussian noise as proposed by Geyer et al. (2017). The results as shown in Figure 3 indicate that the standard trade-off between model quality and privacy holds for FEDDC as well. Moreover, for mild privacy settings the model quality does not decrease. That is, FEDDC is able to robustly predict even under differential privacy. We provide an extended discussion on the privacy aspects of FEDDC in App. A.7.

## 7 EXPERIMENTS ON DEEP LEARNING

Our approach FEDDC, both provably and empirically, improves model quality when using Radon points as aggregation which, however, require convex problems. For non-convex problems, in particular deep learning, averaging is the state-of-the-art aggregation operator. We, hence, evaluate FEDDC with averaging against the state of the art in federated learning on synthetic and real world data using neural networks. As baselines, we consider federated averaging (FEDAVG) (McMahan et al., 2017) with optimal communication, FEDAVG with equal communication as FEDDC, and simple daisy-chaining without aggregation. We further consider the 4 state-of-the-art methods FEDPROX (Li et al., 2020a), FEDADAGRAD, FEDYOGI, and FEDADAM (Reddi et al., 2020). As datasets we consider a synthetic classification dataset, image classification in CIFAR10 (Krizhevsky, 2009), and two real medical datasets: MRI scans for brain tumors,[2] and chest X-rays for pneumonia[3]. We provide additional results on MNIST in App. A.3.8. Details on the experimental setup are in App. A.1.1,A.1.2, code is publicly available at `https://github.com/kampmichael/FedDC`.

**Synthetic Data:** We first investigate the potential of FEDDC on a synthetic binary classification dataset generated by the sklearn (Pedregosa et al., 2011) `make_classification` function with 100 features. On this dataset, we train a simple fully connected neural network with 3 hidden layers on $m = 50$ clients with $n = 10$ samples per client. We compare FEDDC with daisy-chaining period $d = 1$ and aggregation period $b = 200$ to FEDAVG with the same amount of communication $b = 1$ and the same averaging period $b = 200$. The results presented in Fig. 4 show that FEDDC achieves a test accuracy of $0.89$. This is comparable to centralized training on all data which achieves a test accuracy of $0.88$. It substantially outperforms both FEDAVG setups, which result in an accuracy of $0.80$ and $0.76$. Investigating the training of local models between aggregation periods reveals that the main issue of FEDAVG is overfitting of local clients, where FEDAVG *train* accuracy reaches $1.0$ quickly after each averaging step. With these promising results on vanilla neural networks, we next turn to real-world image classification problems typically solved with CNNs.

**CIFAR10:** As a first challenge for image classification, we consider the well-known CIFAR10 image benchmark. We first investigate the effect of the aggregation period $b$ on FEDDC and FEDAVG, separately optimizing for an optimal period for both methods. We use a setting of 250 clients with

---

[2]kaggle.com/navoneel/brain-mri-images-for-brain-tumor-detection

[3]kaggle.com/praveengovi/coronahack-chest-xraydataset

a small version of ResNet, and 64 local samples each, which simulates our small sample setting, drawn at random without replacement (details in App. A.1.2). We report the results in Figure 5 and set the period for FEDDC to $b = 10$, and consider federated averaging with periods of both $b = 1$ (equivalent communication to FEDDC with $d = 1, b = 10$) and $b = 10$ (less communication than FEDDC by a factor of 10) for all subsequent experiments.

Next, we consider a subset of 9600 samples spread across 150 clients (i.e. 64 samples per client), which corresponds to our small sample setting. Now, each client is equipped with a larger, untrained ResNet18.[4] Note that the combined amount of examples is only one fifth of the original training data, hence we cannot expect typical CIFAR10 performance. To obtain a gold standard for comparison, we run centralized learning CENTRAL, separately optimizing its hyperparameters, yielding an accuracy of around 0.65. All results are reported in Table 1, where we report FEDAVG with $b = 1$ and $b = 10$, as these were the best performing settings and $b = 1$ corresponds to equal amounts

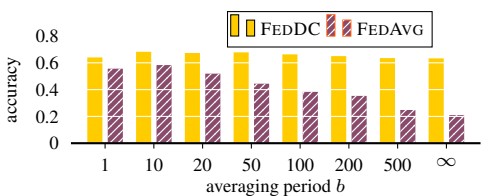

Figure 5: *Averaging periods on CIFAR10.* For 150 clients with small ResNets and 64 samples per client, we visualize the test accuracy (higher is better) of FEDDC and FEDAVG for different aggregation periods $b$.

of communication as FEDDC. We use a daisy chaining period of $d = 1$ for FEDDC throughout all experiments for consistency, and provide results for larger daisy chaining periods in App. A.3.5, which, depending on the data distribution, might be favorable. We observe that FEDDC achieves substantially higher accuracy over the baseline set by federated averaging. In App. A.3.7 we show that this holds also for client subsampling. Upon further inspection, we see that FEDAVG drastically overfits, achieving training accuracies of 0.97 (App. A.3.1), a similar trend as on the synthetic data before. Daisy-chaining alone, apart from privacy issues, also performs worse than FEDDC. Intriguingly, also the state of the art shows similar trends. FEDPROX, run with optimal $b = 10$ and $\mu = 0.1$, only achieves an accuracy of 0.51 and FEDADAGRAD, FEDYOGI, and FEDADAM show even worse performance of around 0.22, 0.31, and 0.34, respectively. While applied successfully on large-scale data, these methods seem to have shortcomings when it comes to small sample regimes.

To model different data distributions across clients that could occur in for example our healthcare setting, we ran further experiments on simulated non-iid data, gradually increasing the locally available data, as well as on non-privacy preserving decentralized learning. We investigate the effect of non-iid data on FEDDC by studying the "pathological non-IID partition of the data" (McMahan et al., 2017). Here, each client only sees examples from 2 out of the 10 classes of CIFAR10. We again use a subset of the dataset. The results in Tab. 2 show that FEDDC outperforms FEDAVG by a wide margin. It also outperforms FEDPROX, a method specialized on heterogeneous datasets in our considered small sample setting. For a similar training setup as before, we show results for gradually increasing local datasets in App. A.3.4. Most notably, FEDDC outperforms FEDAVG even with 150 samples locally. Only when the full CIFAR10 dataset is distributed across the clients, FEDAVG is on par with FEDDC (see App. Fig. 7). We also compare with distributed training through gradient sharing (App. A.3.3), which discards any privacy concerns, implemented by mini-batch SGD with parameter settings corresponding to our federated setup as well as a separately optimized version. The results show that such an approach is outperformed by both FEDAVG as well as FEDDC, which is in line with previous findings and emphasize the importance of model aggregation.

As a final experiment on CIFAR10, we consider daisy-chaining with different combinations of aggregation methods, and hence its ability to serve as a building block that can be combined with other federated learning approaches. In particular, we consider the same setting as before and combine FEDPROX with daisy chaining. The results, reported in Tab. 2, show that this combination is not only successful, but also outperforms all others in terms of accuracy.

**Medical image data:** Finally, we consider two real medical image datasets representing actual health related machine learning tasks, which are naturally of small sample size. For the brain MRI scans, we simulate 25 clients (e.g., hospitals) with 8 samples each. Each client is equipped with a CNN

---

[4]Due to hardware restrictions we are limited to training 150 ResNets, hence 9600 samples across 150 clients.

|              | CIFAR10           | MRI               | Pneumonia         |
| ------------ | ----------------- | ----------------- | ----------------- |
| FEDDC (ours) | **62.9** ±**0.02** | **78.4** ±**0.61** | **83.2** ±**0.84** |
| DC (baseline) | 58.4 ±0.85       | 57.7 ±1.57        | 79.8 ±0.99        |
| FEDAVG (b=1) | 55.8 ±0.78        | 74.1 ±1.68        | 80.1 ±1.53        |
| FEDAVG (b=10) | 48.7 ±0.87       | 75.6 ±1.18        | 79.4 ±1.11        |
| FEDPROX      | 51.1 ±0.80        | 76.5 ±0.50        | 80.0 ±0.36        |
| FEDADAGRAD   | 21.8 ±0.01        | 45.7 ±1.25        | 62.5 ±0.01        |
| FEDYOGI      | 31.4 ±4.37        | 71.3 ±1.62        | 77.6 ±0.64        |
| FEDADAM      | 34.0 ±0.23        | 73.8 ±1.98        | 73.5 ±0.36        |
| CENTRAL      | 65.1 ±1.44        | 82.1 ±1.00        | 84.1 ±3.31        |

Table 1: Results on image data, reported is the average test accuracy of the final model over three runs ($\pm$ denotes maximum deviation from the average).

|                   | CIFAR10           |
| ----------------- | ----------------- |
| FEDDC             | **62.9** ±**0.02** |
| FEDDC +FEDPROX    | **63.2** ±**0.38** |
|                   | **Non-IID**       |
| FEDDC             | **34.2** ±**0.61** |
| FEDAVG (b=1)      | 30.2 ±2.11        |
| FEDAVG (b=10)     | 24.9 ±1.95        |
| FEDPROX           | 32.8 ±0.00        |
| FEDADAGRAD        | 11.7 ±0.00        |
| FEDADAM           | 13.0 ±0.00        |
| FEDYOGI           | 12.5 ±0.04        |

Table 2: Combination of FEDDC with FEDAVG and FEDPROX and non-iid results on CIFAR10.

(see App. A.1.1). The results for brain tumor prediction evaluated on a test set of 53 of these scans are reported in Table 1. Overall, FEDDC performs best among the federated learning approaches and is close to the centralized model. Whereas FEDPROX performed comparably poorly on CIFAR10, it now outperforms FEDAVG. Similar to before, we observe a considerable margin between all competing methods and FEDDC. To investigate the effect of skewed distributions of sample sizes across clients, such as smaller hospitals having less data than larger ones, we provide additional experiments in App. A.3.5. The key insight is that also in these settings, FEDDC outperforms FEDAVG considerably, and is close to its performance on the unskewed datasets.

For the pneumonia dataset, we simulate 150 clients training ResNet18 (see App. A.1.1) with 8 samples per client, the hold out test set are 624 images. The results, reported in Table 1, show similar trends as for the other datasets, with FEDDC outperforming all baselines and the state of the art, and being within the performance of the centrally trained model. Moreover it highlights that FEDDC enables us to train a ResNet18 to high accuracy with as little as 8 samples per client.

## 8 DISCUSSION AND CONCLUSION

We propose to combine daisy-chaining and aggregation to effectively learn high quality models in a federated setting where only little data is available locally. We formally prove convergence of our approach FEDDC, and for convex settings provide PAC-like generalization guarantees when aggregating by iterated Radon points. Empirical results on the SUSY benchmark underline these theoretical guarantees, with FEDDC matching the performance of centralized learning. Extensive empirical evaluation shows that the proposed combination of daisy-chaining and aggregation enables federated learning from small datasets in practice. When using averaging, we improve upon the state of the art for federated deep learning by a large margin for the considered small sample settings. Last but not least, we show that daisy-chaining is not restricted to FEDDC, but can be straight-forwardly included in FEDAVG, Radon machines, and FEDPROX as a building block, too.

FEDDC permits differential privacy mechanisms that introduce noise on model parameters, offering protection against membership inference, poisoning and backdoor attacks. Through the random permutations in daisy-chaining rounds, FEDDC is also robust against reconstruction attacks. Through the daisy-chaining rounds, we see a linear increase in communication. As we are primarily interested in healthcare applications, where communication is not a bottleneck, such an increase in communication is negligible. Importantly, FEDDC outperforms FEDAVG in practice also when both use the same amount of communication. Improving the communication efficiency considering settings where bandwidth is limited, e.g., model training on mobile devices, would make for engaging future work.

We conclude that daisy-chaining lends itself as a simple, yet effective building block to improve federated learning, complementing existing work to extend to settings where little data is available per client. FEDDC, thus, might offer a solution to the open problem of federated learning in healthcare, where very few, undisclosable samples are available at each site.

ACKNOWLEDGMENTS

The authors thank Sebastian U. Stich for his detailed comments on an earlier draft. Michael Kamp received support from the Cancer Research Center Cologne Essen (CCCE). Jonas Fischer is supported by a grant from the US National Cancer Institute (R35CA220523).

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
