# OpenReview forum: "Federated Learning from Small Datasets"
_ICLR.cc/2023/Conference — ICLR 2023 poster_

### Official Review · Reviewer_9XFM · 2022-10-24

**Confidence:** 4
**Correctness:** 4
**Technical Novelty And Significance:** 3
**Empirical Novelty And Significance:** 2
**Recommendation:** 6

**Clarity, Quality, Novelty And Reproducibility:**

The paper is written clearly, it is relatively original, and it seems to be reproducible (code was provided).

**Strength And Weaknesses:**

Strength:
* Interesting paper with a novel method for sharing models in FL systems.
* Seems to work well compared to baseline methods, especially when the number of samples per client is low.
* The theoretical guarantees are a nice addition; although they feel a bit contrived.

Weaknesses:
* Although the communication cost of both FedAvg and FEDDC is linear in the number of clients, in FedAvg we usually sample a subset of the clients while in FEDDC at each round all clients participate. So in practice, FEDDC is substantially more expensive than FedAvg and its variants (and not only by a slight increase).
* As the authors state, their method is susceptible to a new kind of attack which is a malicious client. I admire the authors for being honest about that, nevertheless, this is still a major concern in FL systems.
* The authors claim to compare against SoTA methods, but all baseline methods are obsolete (most recent ones are from 2020). I believe that this is not nearly enough to showcase the authors' claim. In particular, the authors should compare to other methods that were designed for small datasets as well.

**Summary Of The Paper:**

The authors propose FEDDC, a new scheme for training federated learning systems. The main idea is that at each communication round a client can either receive the model of another random client (daisy chaining round) or an aggregated model (aggregation round). The authors show convergence rate and generalization bound for their method in the convex case.

**Summary Of The Review:**

This is a nice paper with interesting approach, yet I think that the comparison to baseline method is severely lacking.

---

### Official Review · Reviewer_L65q · 2022-10-24

**Confidence:** 4
**Correctness:** 3
**Technical Novelty And Significance:** 3
**Empirical Novelty And Significance:** 3
**Recommendation:** 6

**Clarity, Quality, Novelty And Reproducibility:**

The paper has developed an interesting theory that is well substantiated into an implementable algorithm that fixes the poor performance of FedAvg (and its variants) on experiment regimes where local datasets are small. The theoretic insight and its induced algorithm are definitely novel but the theory is restricted to convex loss -- would be good to have a paragraph or appendix section to discuss this.

One simple solution to bad model aggregation is to not forming local model. Local clients can send in gradients so that the server can aggregate them (instead of local model artifacts). Doing so is equivalent to training a full model but it likely converges slower, which might not be critical in low-data regime so please do include comparison with such baseline in the revised version.

In fact, I am a bit curious here: Do we expect the proposed approach to outperform such baseline? If yes, does that mean this approach would perform better than a centralized model? If no, what are scenarios when this method is preferred to such baseline (e.g. is it provably more secure or is it provably faster in convergence even in low-data regime)

The authors pointed the audience at A.1.5 for extra results of this but I went there and did not see relevant results (there are only some other comparisons with FedAvg) -- maybe I missed something?

--

Other questions:

I do not follow this statement:

"Yurochkin et al. (2019) propose local data augmentation for federated learning, but their method requires a sufficient quality of the local model for augmentation which is the opposite of the scenario we are considering"

This is not at all the proposal in that work which does not aim to aggregate local models directly. Instead, local neurons are extracted from local models and are clustered based on whether the neurons encode similar feature aspects; each cluster induces a global neuron which is (loosely speaking) a weighted aggregation of the local neurons. This is how global models are formed in (Yurochkin et al. 2019) which is evaluated under different data configuration for local models (homogeneous or heterogeneous distribution) so this is in fact in the exact same direction of what the authors are considering & should actually be compared with.

**Strength And Weaknesses:**

Strengths:

+ The paper is very well-written -- I like how the theory is presented in a very organized manner which makes it comfortable to follow
+ The solution is novel and opens up a new different perspective for FL with low client data
+ There are rigorous analysis of the proposed idea formalized in forms of concrete theoretical results

Weaknesses:

- Lack of comparison with variants that communicate gradients instead of model updates -- more on this below
- Lack of comparison with baselines in the full-data regime (in addition to what has been demonstrated in low-data regime)
- Developed theory is restricted to convex loss -- would be good to discuss directions to expand towards more general losses -- i think unless the model is linear, the loss will not be convex in the model parameter
- Lack of intuition on what are the advantages Radon point & Radon machine have over a simple weighted average for the aggregation step (other than that the former elicit interesting theoretical results)

**Summary Of The Paper:**

This paper points out the negative impact of small client data on FedAvg: small data results in poor local models; and aggregating with poor local models will impact the performance of FedAvg.

To mitigate this impact, the authors introduce a number of model permutation steps between two model aggregation steps & defining the aggregation operator as computing the Radon point of the set of local models. Before the local models are aggregated, we pair them with a permuted list of clients and send each model to its corresponding paired-up client so that after a few of such permuted redistribution, a local model can draw benefit from multiple local datasets instead of one.

The paper also provides an interesting theoretical analysis which explicitly details the minimum interval between two consecutive aggregation steps (in terms of the sample complexity of the local learning algorithm & the interval between two consecutive permutation steps) -- see Lemma 4 & Proposition 5 -- so that the chance that the generalized error of the aggregated model exceeds a user-specified threshold is doubly-exponentially smaller (see Eq. 2) than that of any local models.

The theory also points out that without the permutation step, the above will not happen if the size of local datasets is smaller than the sample complexity of the local learning algorithm (see Corollary 3).

The empirical results indeed show that the proposed method perform better than other baselines in several low-data simulations derived from several benchmark datasets such as CIFAR10, MRI & Pneumonia.

**Summary Of The Review:**

This is an interesting paper presenting a good mix of elegant theoretical results & new algorithmic ideas. The idea is very different from what has been explored in the literature. The current form of the paper is acceptable to me if the concerns above can be addressed convincingly. At this moment, I am rating this work at weak accept mainly because of the well-developed theory that could potentially inspire more interesting development in the future; but I still remain a bit doubtful of its practical impact -- see my comments above.

---

### Official Review · Reviewer_WcDr · 2022-10-25

**Confidence:** 4
**Correctness:** 3
**Technical Novelty And Significance:** 2
**Empirical Novelty And Significance:** 2
**Recommendation:** 5

**Clarity, Quality, Novelty And Reproducibility:**

This paper is clearly written. The proposed approach has moderate novelty, providing technically sound solutions for a specific FL scenario.

**Strength And Weaknesses:**

Strengths:

+ tackles a practical & important issue in FL: small datasets
+ provides proofs of convergence property;
+ promising empirical results compared with traditional FL on certain small data settings.


Cons:
- My biggest concern about this work is that it is not tested on the most common scenario with*straggler* users. In both the proposed algorithm and empirical experiments, **all** clients have participated in local training and model aggregation/permutation in each round. This is a too strong assumption.
- Under the straggler user scenario, the efficiency of 'permutating models across clients' would largely drop, and the model performance will theoretically and empirically degrade given the same learning rounds. Authors should provide experiments given different ratios of active users.
- Another untouched issue in this work is data heterogeneity, which could be a more challenging problem than small datasets. In the experiments, each client data distribution is still i.i.d when they are assigned with samples randomly drawn from a global view. It is not surprising that FedDC would perform well given such data distributions. I am curious how FedDC would perform compared with FedAvg/FedProx when there is data heterogeneity among clients.


**Summary Of The Paper:**

This paper proposed to tackle data insufficiency in Federated Learning through daisy-chain training, which interchanges model aggregation and model permutation across clients. A theoretical convergence guarantee is provided. Empirical results show that the proposed algorithm FedDC outperforms FedAvg and other baselines on certain benchmarks with small data samples per client.

**Summary Of The Review:**

This paper tackles data insufficiency in FL by interleaving model aggregation and permutation. The approach outperforms traditional FL on small dataset scenarios, although the approach is not tested on more practical cases, such as FL with straggler users or with data heterogeneity.

---

### Official Review · Reviewer_LWYx · 2022-10-27

**Confidence:** 3
**Correctness:** 3
**Technical Novelty And Significance:** 3
**Empirical Novelty And Significance:** 3
**Recommendation:** 6

**Clarity, Quality, Novelty And Reproducibility:**

The paper is very well written and clear to follow. The paper is very well rounded both in terms of theory and empirical results. The approach is quite original.

**Strength And Weaknesses:**

Strengths:
+ The paper is very well written and the problem under consideration is super well motivated.
+ While device selection is not taken into account, the authors restrict their attention to cross-silo settings, where approaches which involve full device participation scale well. The theoretical results are comprehensive in terms of convergence, coverage in terms of local datasets and model improvement.
+ The experimental results are also comprehensive in terms of various aspects of non-iidness and beats considered baselines consistently.
+ One of the weaknesses of FedDC because of daisy chaining is because of each device having an opportunity of reconstructing data from model updates from other devices. The authors acknowledge the shortcoming and propose to address it through the usage of gradient clipping and Gaussian mechanism.

Weaknesses:
- Inspite of using DP related techniques, the exposure of model updates to each individual device allows a malicious client to retain the model and engage in a data reconstruction attack and subsequently do so for each client effectively. It is not clear what level of Gaussian mechanism would be able to alleviate such a privacy loss and what would be the effect of such a scheme on the performance of the algorithm.
- The other concern is regarding experiments where in each empirical evaluation the number of samples across clients is the same. Its not clear as to what would be the impact of daisy chaining, when encountering clients which have fewer samples. None of the experiments in the paper seem to study that.


**Summary Of The Paper:**

This paper considers the interesting problem of cross-silo FL with very small datasets, with some devices having only two samples per device. With an aptly crafted aggregation scheme, the authors propose to interleave model aggregation and permutation steps, which lets every model to be trained through multiple local datasets before aggregation. For the proposed algorithm FedDC, strong theoretical bounds are derived. Finally experimental results demonstrate the usefulness of the approach.

**Summary Of The Review:**

The cross-silo setting in FL with devices containing small number of samples is a problem of particular interest. The authors propose FedDC and provide a well rounded justification in terms of theoretical and experimental evaluation. The paper is in a pretty good shape. Adding some experimental evidence in terms of dealing with devices with unequal number of samples would only strengthen the paper.

---

### Official Review · Reviewer_mrq4 · 2022-10-29

**Confidence:** 3
**Correctness:** 2
**Technical Novelty And Significance:** 2
**Empirical Novelty And Significance:** Not applicable
**Recommendation:** 5

**Clarity, Quality, Novelty And Reproducibility:**

Clarity: this paper proposes a daisy-chaining based FL learning framework, but the algorithm is not presented clearly. Oracle agg means performing average, but fig.1 (rirght) seems allowing partial average (note that $\bar{h}$ still needed here). The proofs are not derived in a step-by-step way. For example, most of the convergence analysis used other papers' result directly.

Quality: the idea of daisy chaining is interesting and it is expected that FedDC would work well and outperform others.  The theoretical analysis is not rigorous, where there is a gap between the high level design of the daisy chaining and implemented algorithm.

Novelty: However, actually this strategy is very similar as permutation based SGD training by shuffling the data while here FedDC reshuffles the clients.

Reproducibility: the code is publicly accessible but not runnable.


**Strength And Weaknesses:**

Strength: this work considers a practical issue of data sparse in the FL systems. It further provides theoretical analysis results on convergence, generalization performance evaluation, requirements on communications, and massive numerical results on classification problems.

Weaknesses: even the work is new and encouraging, there are still some major concerns as follows:

1) A convergence result is given in Corollary 1, which basically uses Yu et al. (2019)' result. How does the permutation affect the convergence is not analyzed. Does Yu et al. (2019)'s analysis covers the client permutation? Note that there is a significant difference between the selection with and without replacement. There is a gap between the FedDC and the analysis.

2)  In the convergence result, $b$ needs to be less than a threshold, but eq.(2) $b$ is required to be large than a threshold. Is there an overlap between these?

3) This work assumes that the risk considered is convex while their convergence result is borrowed from the nonconvex. A consistent argument is encouraged.

4)  The authors claimed that FedDC can perform better in the data sparse case, however, the theory seems uncorrelated with any parameters regarding the data sample size (Lemma 4). In Prop. 5, if m is small and n is large, will $b$ be a negative number? A sufficient discussion of the theoretical results should be provided.

5) $m$ is assumed to be large enough so that the local model can improve the performance. Will this method also work for $m$ is small in the sense that the same as FedAvg or will diverge? also, how decide $r$ in practice?

6) The theoretical results seem completely independent on which federated learning algorithm adopted, proxFed, or adam based algorithm. In other word, the convergence rate of the algorithm will not affect the $b$ and $T$, right?




**Summary Of The Paper:**

In this work, a practical and efficient FL learning framework is proposed for improving aggregation performance over multiple clients. The core step of this model is to use the interleaving model aggregation and permutation steps so that the proposed federated daisy-chaining method can work well for data sparse problems. Multiple experimental results show that FedDC outperforms the state-of-the-art methods significantly, e.g., FedAvg.

**Summary Of The Review:**

In summary, this paper proposes an efficient training framework, FedDC, for dealing with the data sparse problem, but the theoretical justification of the convergence and generalization contains a gap, which weakens the contributions of this work.



====================== after Zoom Meeting ===================

After the discussion with the AC and other reviewers, I suppose that there are some merits of this work and increase the score, even 1) the setting of the convergence part is not consistent with the PAC analysis and there is no condition of step size to ensure the convergence; 2) tuning the multiple hyper-parameters to satisfy the theoretical conditions is a very difficult task, 3) and there is still a gap between the theory and practical implementation for the PAC part.

---

### Author Response · Authors · 2022-11-18
**Rebuttal summary**

Dear reviewers and AC,

We would like to thank everyone for their constructive feedback. The discussion phase has allowed us to clarify and answer questions of reviewers that helped us to improve our paper. In particular, we
- improved the presentation to address reviewer's questions,
- discussed the parameters of our theoretical analysis,
- clarified reviewers’ requests that were already addressed in the appendix (full-data regime, non-iid data, different local dataset sizes, sending gradients instead of models), and
- ran and included additional experiments on straggler uses/ limited amount of client participation.

We are grateful for the time and effort put in by the reviewers in evaluating this work and are confident that our rebuttal addresses the open concerns and questions of the reviewers. We kindly ask you to reconsider your score based on our answers.

Sincerely,

The authors

---

### Decision · Program_Chairs · 2023-01-20

**Decision:**

Accept: poster

**Justification For Why Not Higher Score:**

There are still numerous concerns.

1. There is a theory gap between convex with radon points, convex with averaging, non-convex with averaging in convergence and PAC-like generalization guarantee. Also, it is not clear from the proof and analysis on how to choose step size and hyper-parameters to make it both theoretical satisfactory and practically useful.

2. Experiments need to be stronger.  Some of the baselines need to be SOTA.

3. One reviewer pointed out that, from the DP perspective, permutation can mitigate malicious attacks to some degree but it will not totally avoid it.  Data privacy may still be an issue under the FedDC setting.


**Justification For Why Not Lower Score:**

The idea is interesting and the work is sufficiently novel for the conference. Despite the above lingering concerns the work may have its value and impact to the federated learning community.

**Metareview: Summary, Strengths And Weaknesses:**

In this paper the authors propose a federated daisy chaining (FedDC) strategy to deal with small local datasets in federated learning.  Small local datasets may result in poor local models.  Daisy chaining can mitigate this issue where models are trained locally one after another.  But daisy chaining may result in privacy issue.  The proposed FedDC strategy introduces a permutation step in the conventional daisy chaining to improve the protection of data privacy. This gives rise to an interleaving pattern between aggregation and permutation.  The authors show the convergence of FedDC for convex functions and provide PAC-like generalization guarantees when the aggregation is conducted using Radon points.  Experiments are carried out on non-convex deep neural networks to compare with numerous existing federated learning settings under the small local dataset scenario.  The authors also put up a good rebuttal to address the concerns raised by the reviewers.   That being said, the following concerns still stand after a discussion between the AC and the reviewers.

1. There is a theory gap between convex with radon points, convex with averaging, non-convex with averaging in convergence and PAC-like generalization guarantee. Also, it is not clear from the proof and analysis on how to choose step size and hyper-parameters to make it both theoretical satisfactory and practically useful.

2. Experiments need to be stronger.  Some of the baselines need to be SOTA.

3. One reviewer pointed out that, from the DP perspective, permutation can mitigate malicious attacks to some degree but it will not totally avoid it.  Data privacy may still be an issue under the FedDC setting.

Overall,  all reviewers consider the idea interesting and the work technically novel enough for this conference.  The experimental results are supportive.  The proposed FedDC may have its value to the federated learning community.

**Note From Pc:**

if the above contains the word "oral" or "spotlight" please see: "oral" presentation means -> notable-top-5% and "spotlight" means -> notable-top-25%. As stated in our emails, we are disassociating presentation type from AC recommendations

**Summary Of Ac-Reviewer Meeting:**

A virtual meeting took place between AC and 4 of the 5 reviewers.

1. All reviewers consider the idea interesting and the work is technically novel enough for this conference.

2. There is a theory gap between convex with radon points, convex with averaging, non-convex with averaging in convergence and PAC-like generalization guarantee. Also, it is not clear from the proof and analysis on how to choose step size and hyper-parameters to make it both theoretical satisfactory and practically useful.

3. Experiments need to be stronger.  Some of the baselines need to be SOTA.

4. One reviewer pointed out that, from the DP perspective, permutation can mitigate malicious attacks to some degree but it will not totally avoid it.  Data privacy may still be an issue under the FedDC setting.

The reviewers who participated in the meeting think that despite the above lingering concerns the work may have its value and impact to the federated learning community.